# Integrating Remote Photoplethysmography and Machine Learning on Multimodal Dataset for Noninvasive Heart Rate Monitoring

**DOI:** 10.3390/s24237537

**Published:** 2024-11-26

**Authors:** Rinaldi Anwar Buyung, Alhadi Bustamam, Muhammad Remzy Syah Ramazhan

**Affiliations:** 1Department of Mathematics, Faculty of Mathematics and Natural Science, Universitas Indonesia, Depok 16424, Indonesia; rinaldi33@ui.ac.id (R.A.B.); muhammad.remzy@sci.ui.ac.id (M.R.S.R.); 2Data Science Center (DSC), Faculty of Mathematics and Natural Science, Universitas Indonesia, Depok 16424, Indonesia

**Keywords:** heart rate, machine learning, remote photoplethysmography, signal processing

## Abstract

Non-contact heart monitoring is crucial in advancing telemedicine, fitness tracking, and mass screening. Remote photoplethysmography (rPPG) is a non-contact technique to obtain information about heart pulse by analyzing the changes in the light intensity reflected or absorbed by the skin during the blood circulation cycle. However, this technique is sensitive to environmental lightning and different skin pigmentation, resulting in unreliable results. This research presents a multimodal approach to non-contact heart rate estimation by combining facial video and physical attributes, including age, gender, weight, height, and body mass index (BMI). For this purpose, we collected local datasets from 60 individuals containing a 1 min facial video and physical attributes such as age, gender, weight, and height, and we derived the BMI variable from the weight and height. We compare the performance of two machine learning models, support vector regression (SVR) and random forest regression on the multimodal dataset. The experimental results demonstrate that incorporating a multimodal approach enhances model performance, with the random forest model achieving superior results, yielding a mean absolute error (MAE) of 3.057 bpm, a root mean squared error (RMSE) of 10.532 bpm, and a mean absolute percentage error (MAPE) of 4.2% that outperforms the state-of-the-art rPPG methods. These findings highlight the potential for interpretable, non-contact, real-time heart rate measurement systems to contribute effectively to applications in telemedicine and mass screening.

## 1. Introduction

Heart rate is one of the most important vital signs observed to determine a person’s health. This is because heart rate directly indicates cardiovascular diseases (CVDs) such as heart disease, arrhythmia, stroke, and high blood pressure. CVD is a noncommunicable condition that can lead to sudden death due to issues with the heart and blood vessels [1,2]. In addition, heart rate can also be an indicator of physical and emotional stress. The precious information provided by the heart rate makes it important to monitor. There are some tools for heart rate monitoring, such as stethoscopes, electrocardiograms, and pulse oximeters. However, those methods often involve uncomfortable physical contact with sensors, presenting challenges in continuous monitoring and real-time data collection [3,4,5].

The first non-contact heart rate monitoring techniques using optical sensors were invented by Hertzman in 1937, developing with the advancement of science and technology in medicine, known as photoplethysmography (PPG) [6]. PPG allows the measurement of the heart rate by placing a sensor directly on the skin, typically at areas like the fingertip or earlobe, to detect changes in the light absorbed or reflected by the skin due to blood volume pulsatile. With the development of light source technology and photo detectors in the 1960s, more accurate and portable PPG devices were feasible. With further advances in electronics and signal processing in the 1970s, applications of PPG became more feasible in a clinical environment, as devices could now be smaller and more integrated.

In the early 2000s, researchers began exploring the potential of detecting PPG signals remotely using regular digital cameras which later provided the roots for remote photoplethysmography (rPPG). The main idea of rPPG is that skin color changes from the circulating blood flow were to be recorded and analyzed, mainly on facial regions. Initial attempts demonstrated that small changes in skin color due to blood flow could indeed be captured using webcams and camcorders, paving the way for remote, non-contact measurements. By using this technique, heart rate monitoring can be carried out remotely and faster. One of the rPPG technique implication is mass screening for life insurance underwriting. By using this approach, the assessment of prospective policyholders is much faster and cheaper [7,8,9].

Different algorithms have been developed to perform heart rate calculations using the rPPG technique. The first algorithm developed is called Green’s algorithm, presented by Verkruysse et al. [10] in 2008. This algorithm highlights the green channel to find changes in blood volume within cutaneous blood vessels, which can later be used to monitor heart rate. The reason for choosing the green channel is related to the favorable absorption characteristics of hemoglobin. However, it is susceptible to individual differences in skin pigmentation, which might result in changing the obtained signal because of the various amounts of light absorbed and reflected.

In 2010, Poh et al. [11] developed a method for calculating heart rate named Independent Component Analysis (ICA). This method is a statistical method used to separate overlapping signals into independent components. The use of ICA in rPPG is very important because it helps improve the accuracy of noninvasive heart rate monitoring via video, making it very useful in more convenient and easy health monitoring. However, ICA has a major weakness—the assumption that the source signals that want to be separated are completely independent of each other. In practice, especially in the rPPG technique, the signal is related to the heartbeat, and the noise is not always completely independent.

Therefore, the next method is proposed, namely chrominance-based (CHROM) rPPG signal extraction proposed by De Haan and Jeanne [12] in 2013. CHROM does not completely rely on the assumption of independence. Instead, CHROM utilizes color relationships in the video to separate the heartbeat signal from noise so that it is more robust to situations where the signal is not completely independent. Although more stable than ICA, CHROM can still be affected by drastic changes in lighting because these changes affect the chrominance values used. Therefore, this method was refined by research conducted by W. Wang et al. [13] in 2017, who proposed the algorithm plane orthogonal to skin (POS). Due to the linear combination of color signals in the POS representation, it is invariant to the changes in the lighting conditions and, correspondingly, reduces the intensity variations within an improved signal stability.

Recently, with the emergence of deep learning, many studies on noninvasive heart rate measurement are shifted towards deep learning models such as convolutional neural network (CNN) [14,15,16,17]. However, CNNs often work as black boxes, making it difficult to understand how the inputs lead to the outputs. The lack of interpretability may raise concerns about accountability and reliability in many applications, especially in healthcare. On the other hand, rPPG has an advantage over CNN in terms of interpretability since the theoretical background of rPPG is rooted in optics and physiology, not to mention that the rPPG mechanisms are straightforward, making it easier for practitioners to understand how the measure correlates with physiological signal [18].

Although progress has been made in rPPG for contactless heart rate measurement, the current methods still struggle with noise and other external influences, including lighting conditions and the skin pigmentation of the individual. Such challenges hint that the current methods need a more wholesome approach by incorporating machine learning integration and multiple data types to enhance robustness and accuracy. In the event of these gaps, our research proposes incorporating integration machine learning and a multimodal approach: 1 min facial video data along with physical characteristics, such as age, gender, height, weight, and body mass index (BMI), to enhance heart rate estimation. Two machine learning algorithms, namely support vector regression (SVR) and random forest, will be utilized to model the data input to predict the heart rate. Ablation experiments will be employed by removing or replacing each modality to investigate the impact on the model prediction. Our work will contribute to the integration of robust heart rate measurement for more personalized estimation, considering variability between individuals.

## 2. Methodology

### 2.1. Hypothesis

A study published by Quer et al. [19] utilized wearable fitness trackers to investigate age, body mass index, and sex-related differences in resting heart rate among 92,457 adults. It was concluded that the association of age and body mass index with RHR were statistically significant but small. Overall, the RHR tends to decrease with increasing age and was slightly higher with increased BMI. Similarly, two studies conducted by Ehrenwald et al. [20] and Gonzales et al. [21] studied the temporal changes in RHR. It has been observed that in general, the RHR decreases as age increases and women have a higher RHR compared to men. Furthermore, changes in BMI also affect the RHR; an increase in body mass results in an increased heart rate. Consequently, based on this study, we hypothesized that the inclusion of the rPPG technique and physical attributes, such as age, gender, weight, height, and BMI, may enhance the measurement system in its performance and reliability aspects.

### 2.2. Remote Photoplethysmography

Remote photoplethysmography (rPPG) is an innovative technique that allows measuring the heart rate noninvasively and without direct contact with the skin. The basic principle involves using light emitted onto the surface of the skin and detecting changes in the intensity of light reflected or absorbed by the skin during the blood circulation cycle [22]. The resulting signal, known as blood volume pulse, describes changes in blood vessel volume over time and allows the measurement of heart rate. The skin color measured by the camera is a combination of the intensity and spectrum of the light source, the intrinsic skin color, and the sensitivity of the camera’s color channels. Temporal changes in skin color are proportional to the magnitude of lighting intensity that would be influenced by specular variations caused by movement and subtle color changes due to blood circulation. In rPPG, two components contribute to the observed color variations on the skin surface: specular and diffuse reflections.

Specular reflection refers to light reflection due to a smooth surface, such as the skin. This is the result of the direct reflection of light from the skin without any scattering. For many cases, it may be the most dominant component in observed color variations and overshadows other components. However, this does not contain information regarding heart rate, so it needs to be rejected or minimized in the rPPG technique. On the other hand, diffuse reflection may be said to deal with skin tissue absorption and light scattering. This is influenced by factors such as the hemoglobin and melanin content of the skin [23]. Diffuse reflection contains information related to changes in blood volume in the vessels around the skin. This component is of interest in rPPG because it can be used to extract signals related to changes in blood volume [24]. Figure 1 shows an illustration of the use of light emitted into the skin in the rPPG technique.

To extract heartbeat signals from observed color variations, the rPPG method focuses on isolating and analyzing diffuse reflection components while minimizing the impact of specular reflections. For this purpose, the Plane-Orthogonal-to-Skin (POS) approach put out by [13] is the methodology employed in this study. The main idea of the POS is that the video sequence captures a visible skin region, where color variations correspond to heartbeat-related blood flow changes. The POS algorithm aims to account for variations in skin tone by projecting the RGB signal onto a plane orthogonal to the temporally normalized skin tone direction when given a 15 s facial video where each frame’s RGB values are averaged to obtain a single-color intensity value per frame, representing the overall color change over time. This results in a temporal array of $R\left(n\right),\;G\left(n\right),$ and B(n) where n represents frame index. The array is then projected using a matrix $p_{x}=\left(\begin{array}{ccc} 0 & 1 & -1\\ -2 & 1 & 1 \end{array}\right)$, designed to minimize the influence of skin tone on pulse extraction. This guarantees that there are as many beat components as possible in the final rPPG signal. After that, the signal’s component will be separated to calculate heart rate representation. The algorithm of POS is formulated in Algorithm 1 as follows:
**Algorithm 1:** Plane orthogonal to skinAssumption: Video duration 15 s; Frame rate 30 fps; total frame 450 framesInput: Video contains N frameOutput: rPPG signal1:   H = zeros (*l*, *N*), *l =* 48 (30 fps camera)2:   for n=1,2,…, N do: 3:     $C\left(n\right)={\left[R\left(n\right),\;G\left(n\right),\;B(n)\right]}^{T}$4:     if m=n−l+1>0 then5:       $C_{n}^{i}=\frac{C_{m\to n}^{i}}{\;\mu \;C_{m\to n}^{i}}$6:       $S=\;\left(\begin{array}{ccc} 0 & 1 & -1\\ -2 & 1 & 1 \end{array}\right)\cdot C_{n}$
7:       $S_{1}=S\left[1\right]$ # The first row of S
8:       $S_{2}=S\left[2\right]$ # The second row of S
9:       $h=S_{1}+\frac{\sigma \left(S_{1}\right)}{\sigma (S_{2})}\cdot S_{2}$10:        $H_{m\to n}=H_{m\to n}+(h-\mu \left(h\right))$11:    end if12:   end for13:   return H


This algorithm has the benefit of being less constrictive regarding the degree of distortion shown in the video and more resilient against noise. It is appropriate for a range of applications since it can function well in both fixed and moving scenarios.

### 2.3. Signal Filtering

In the actual world, noise frequently taints signals. There are many different types of noise, such as environmental interference, jitter in data acquisition, and electromagnetic interference. To make the signal being measured or analyzed cleaner and more precise, noise can be reduced or eliminated using the filter process. Butterworth bandpass filter, which is thought to enhance the quality of rPPG signals, is one of the most popular filters [25,26]. According to Steiglitz [27], Butterworth filters are made to have a flat frequency response within the passband frequency range, which permits all frequencies to go through without distortion. The transfer function, a polynomial ratio of output to input, is a way the filter process response to input signals may be described. The mathematical expression for the transfer function is as follows:(1)$$H(s)=\frac{1}{s^{2}+\left(2\sqrt{n}\cdot \pi \cdot f_{c}\right)s+N\pi^{2}f_{c}^{2}}$$
where the transfer function H(s) is the transfer characteristics, and s is a complex variable function. The parameter N indicates the filter order, whereas $f_{c}$ signifies the cutoff frequency. In this study, a Butterworth filter will use the cutoff parameters in the range 0.8–2.0 Hz and the order used is 4. The reason for choosing this interval is because the average resting human heart rate is in the interval 48–120 bpm [28]. A fourth-order Butterworth filter is chosen for several reasons. First, higher-order filters provide steeper roll-offs, which enhance the filter’s ability to sharply distinguish between the passband and the stopband.

### 2.4. Fast Fourier Transform

After being filtered, the rPPG signals may now be used to obtain crucial information for heart rate computations. The Fourier transform may be used to analyze frequency by applying it to the rPPG signal. Signals in the time domain will be converted into the frequency domain using this technique. Consider a discrete time signal x[t] of length N (t=0, 1, 2, ..., N−1). Discrete Fourier transform (DFT) from x[t] can be defined as follows:(2)$$X\left[f\right]=\sum_{t=0}^{N-1}x\left[t\right]e^{-j\;\frac{2\pi kt}{N}}$$
where X[f] is the $f_{th}$ frequency spectrum, x[t] is time signal of $t_{th}$ sample, and N is the signal length. However, $O(n^{2})$ operations are needed for classical DFT, which can be quite slow for long signal lengths. This issue is resolved by the fast Fourier transform (FFT), which divides the DFT into smaller DFTs that can be computed more quickly. The Cooley–Tukey algorithm is the most well-known FFT algorithm. The FFT is split into two smaller pieces by this algorithm, which then processes each piece recursively [29].

### 2.5. Support Vector Machine

The support vector machine is one of the most powerful models of machine learning widely used for both classification and regression. SVMs which are used for regression tasks are called support vector regression (SVR). It has based its idea on finding the optimal decision boundaries or a hyperplane with maximum margin [30]. The main optimization problem of the hyperplane for regression involves the minimization of the sum of slack variables, both positive and negative, allowing deviations within a margin ϵ from the true target value. Let N be the total number of data points in the training set; then, the formula is as follows:(3)$$\begin{array}{l} \underset{w,b}{\min}\frac{1}{2}{\left\|w\right\|}^{2}+C\sum_{n=1}^{N}{(\xi }_{n}+\hat{\xi_{n}})\\ \;s.t\;t_{n}\leq y\left(x_{n}\right)+\epsilon +\xi_{n}\\ \;t_{n}\geq y\left(x_{n}\right)-\epsilon -\hat{\xi_{n}}\\ \;\xi_{n},\;\hat{\xi_{n}}\geq 0 \end{array}$$

The w defines the weight vector that sets the hyperplane separating the classes in a higher-dimensional space. Therefore, ${\left\|w\right\|}^{2}$ is the squared norm of the weight vector, which represents the margin of separation between classes. Variable C is a regularization parameter that controls the trade-off between maximizing margin and minimizing errors. Two slack variables $\xi_{n}$ and $\hat{\xi_{n}}$ are introduced to handle deviations above and below the margin. The Figure 2 below illustrates the hyperplane problem of the SVR and the role of each variable.

However, Equation (3) is complicated to solve. Therefore, the dual form will be formulated using the Lagrange function and Karush–Kuhn–Tucker (KKT) condition. For regression problems, the dual of the hyperplane optimization method can be obtained using similar techniques. Therefore, the dual of regression hyperplane optimization is given as follows:(4)$$\begin{array}{l} \underset{a,\;\bar{a}}{\mathrm{argmax}}-\frac{1}{2}\sum_{n=1}^{N}\sum_{m=1}^{N}{(a}_{n}-{\bar{a}}_{n})\;(a_{m}-{\bar{a}}_{m})k\left(x_{n},x_{m}\right)-\epsilon \sum_{n=1}^{N}a_{n}+{\bar{a}}_{n}\\ \;+\sum_{n=1}^{N}\left(a_{n}+{\bar{a}}_{n}\right)t_{n}\\ \;s.t\;0\leq a_{n}\leq C,\;0\leq {\bar{a}}_{n}\leq C\\ \;\sum_{n=1}^{N}(a_{n}-{\bar{a}}_{n})=0 \end{array}$$
where $a_{n}$ is associated with the upper constraint $t_{n}\leq y\left(x_{n}\right)+\epsilon +\xi_{n}$ and ${\bar{a}}_{n}$ is associated with the lower constraint $t_{n}\geq y\left(x_{n}\right)-\epsilon -\hat{\xi_{n}}$. Variable $k\left(x_{n},x_{m}\right)$ is a kernel function. There are several kernel functions such as linear, polynomial, and RBF (Radial Basis Function) kernel [31].

The SVR model can be implemented for multimodal heart rate prediction by incorporating signal features derived from photoplethysmography alongside the physical information, such as age and body mass index, to improve accuracy. The SVR will build the best model based on the optimization of the hyperplane problem during the training process. The best model represented in mathematical function will be used to predict the heart rate based on the input. To enhance machine learning performance, the hyperparameters of the SVR need to be tuned. One of the most important hyperparameters to be tuned is the kernel function. The kernel function converts data into the right format, hence allowing SVRs to identify the best boundaries. Another hyperparameter to be tuned is C which is the balance between attempting to fit the training data with a minimal error and maximizing the margin of the decision boundary; a higher value of C will make the considerations of correct classification more important. γ is a factor associated with the influence of training data on the decision boundary, whereas ϵ is a margin within which no penalty is incurred in the training loss function [32].

### 2.6. Random Forest Regression

The random forest regression is an ensemble machine learning model that relies on the bagging approach: building multiple base models in parallel, and the final prediction is performed with an average vote, ensuring variance reduction by building models using several bootstrap datasets [33]. It consists of several fixed-size base models, which are usually classification and regression trees (CARTs). It builds binary trees, where each internal node splits the data into two child nodes based on a split criterion. The goal is to create a tree structure that maximizes the accuracy of predictions for new data points.

Given the training data $\left\{x_{n},\;t_{n}\right\},\;n=1,\ldots ,n$, where $t_{n}$ is a label. Let Q be the set of data at a node, $\theta =\left(j,a\right)$ be the set of candidate feature j and threshold value a, and $Q_{left}(\theta )$ and $Q_{right}(\theta )$ be the data in the left and right partitions of the threshold, defined as the following equation:(5)$$\begin{array}{cc} Q_{left}\left(\theta \right) & =\left\{\left(x,t\right)|x^{\left(j\right)}\leq a\right\}\\ Q_{right}\left(\theta \right) & =Q-Q_{left}(\theta )\\ G\left(Q,\;\theta \right) & =\frac{N_{left}}{N}H\left(Q_{left}(\theta )\right)+\frac{N_{right}}{N}H\left(Q_{right}(\theta )\right) \end{array}$$
where N is the number of data points at the node, and $N_{left}$, $N_{right}$ are the numbers of data points in the left and right partitions, respectively. Therefore, selecting the feature and threshold that produces the purest partition at a node can be expressed as the following optimization problem:(6)$$\theta^{*}=\underset{\theta }{\min}G\left(Q,\;\theta \right)$$

For regression tasks, the input is divided into several partitions called $R_{m}$. Each region contains a subset of the data points (observations) that fall within it. The goal is to find optimal partitions that minimize the error in each region. The node m represents a region $R_{m}$ with $N_{m}$ observations, and the mean of the target values within each region is $c_{m}$. The common criterion to minimize is the mean squared error $H(x_{M})$ as shown in the following equations.
(7)$$\begin{array}{cc} c_{m} & =\frac{1}{N_{m}}\sum_{i\in N_{m}}t_{i}\\ H\left(X_{m}\right) & =\frac{1}{N_{m}}\sum_{i\in N_{m}}{\left(t_{i}-c_{m}\right)}^{2} \end{array}$$

Next, using the steps above, the random forest regression algorithm builds multiple decision trees by bootstrapping and aggregating the results. The algorithm to perform random forest regression is given in Algorithm 2 as follows:
**Algorithm 2:** Random forest regressionInput: training dataset, *n*: number of trees; *d*: tree depth; *m*: number of features at each nodeOutput: prediction of Random Forest model1:  for *i* = 1, 2, …, *n* do: 2:   Bootstrap sampling of the dataset *D_i_*3:   Build decision tree: 4:   At each node, randomly sample m features5:   Find the best split using the *m* subset6:   Split the node and repeat until stopping criteria (max depth or minimum
samples) are met.
7:   Save the decision tree *T_i_*
8:  end for 9:  For each tree *T_i_*, get the predicted value from all trees10: return Average of the predictions from each tree

The random forest model can be implemented for multimodal heart rate prediction by incorporating signal features derived from photoplethysmography alongside the physical information, such as age and body mass index, to improve accuracy. Multiple decision trees will be built to model the data, and the final prediction will be calculated by averaging the result of each tree.

For random forests, there are three main hyperparameters that will be tuned to improve the model’s performance. Firstly, the number of trees which controls how many trees are involved in the ensemble. Secondly, max depth is the setting that controls the depth of each tree, and this must be performed to avoid overfitting. The final hyperparameter, minimum samples per split denotes the minimal number of samples that are necessary to split an internal node, which prevents the trees from growing too complex [34]. The illustration of the random forest regression can be seen in Figure 3.

### 2.7. Data Collection

In this experiment, a dataset with the facial video recordings of 60 individuals was utilized. Each video’s length was standardized to 1 min, where the resolution and frame rate were set to be uniform at 480×640, and 30 fps, respectively. This resolution and frame rate are used to balance the performance and computational efficiency, leading to an accurate prediction but still feasible for real-time implementation. The data collection was conducted at the same place and with fixed lighting and subjects had to take off any facial accessories, such as eyeglasses. Together with video recording, there was manual measurement using an oximeter to gather the actual value as the ground truth. The height in cm, weight in kg, age in years, and gender are also recorded in the dataset. To ensure privacy, we anonymized the subjects by masking facial features such as eyes, nose, and mouth to prevent unauthorized recognition. We also introduced variable body mass index (BMI) into the dataset. Figure 4 depict the characteristics of the dataset used in this study.

From Figure 4a, it can be concluded that from the 60 individuals, there are 45 males and 15 females in the dataset. Figure 4b shows the variations in heart rate between the males and females. It also shows that females generally present a higher heart rate compared to males, since their median rate is just above 75 beats per minute, while the median for males is just about 70 beats per minute. The interquartile range (IQR), which represents the middle 50% of the dataset, is wider for females, ranging approximately between 70 and 85 bpm. This indicates that there is a greater spread or deviation in the pulse rates of females. In contrast, the IQR for males varies between 65 and 80 beats per minute; this means that male heart rates are more huddled around the lower end of the spectrum. Figure 4c–f indicate that our dataset has a good diversity of heart rate, height, weight, and BMI, making this ideal for training machine learning.

### 2.8. Region of Interest Selection

Region of interest (RoI) selection in the rPPG technique involves selecting a specific area on a person’s skin or face from which the rPPG signal will be extracted. The face was chosen as the RoI in the rPPG algorithm because it has several advantages over other body parts. First, the face has a high concentration of capillaries, making it easier to detect changes in blood flow. Second, faces have a low level of reflection, making it easier to separate PPG signals from environmental noise. Third, faces are easy to access and reach, making them suitable for real-time video capture applications [26,35]. For this purpose, we utilized Dlib facial landmarks that combine Histogram of Oriented Gradients (HOG) as the feature extraction and SVM as the key points classifier [36]. The RoI used in this research is the facial skin without eyes, nose, and lips. For this purpose, the landmarks (36, 39, 40, and 41) for the left eye, (42, 45, 46, and 47) for the right eye, (32, 33, 34, 35, and 36) for the nose, and (49, 50, 51, 52, 53, 54, 55, 56, 57, 58, 59, and 60) for the lips will be filled as black as shown in Figure 5.

### 2.9. Feature Selection

In machine learning, signal feature selection is the process of identifying and selecting the most relevant features included in the signal data that contribute to an improvement in the performance of machine learning models. After several surveys in the literature [27,37,38,39], we decided to utilize signal features as follows:
Maximum and minimum values of the signal: They are the basic features of a signal representing the maximum and minimum value of signal amplitude, respectively. From those values, a new feature called peak to peak, which is the difference between the maximum and minimum values of the signal, can be generated. Given that x(t) represents the signal amplitude at time frame t, the maximum, minimum, and peak-to-peak values of the signal are represented as follows:(8)$$\begin{array}{cc} max_{value} & =\underset{t}{\max}x(t)\\ min_{value} & =\underset{t}{\min}x(t)\\ peak\;to\;peak & =max_{value}-min_{value} \end{array}$$Skewness and Kurtosis: Skewness is a measure of the asymmetry of the signal’s amplitude distribution around the mean, while kurtosis is the existence of outliers in the signal’s distribution. Given that x(t) represents the signal amplitude at time frame t, and μ,σ are the mean and standard deviation of the signal, respectively, skewness and kurtosis of the signal are represented as follows:(9)$$\begin{array}{l} x_{ske}=\frac{E\left({\left(x\left(t\right)-\mu \right)}^{3}\right)}{\sigma^{3}}\\ x_{kur}=\frac{E\left({\left(x\left(t\right)-\mu \right)}^{4}\right)}{\sigma^{4}}-3 \end{array}$$Entropy: It measures the complexity or randomness in the signal based on the probability distribution of its values. It is often used to assess the predictability of a signal. Given that x(t) represents the signal amplitude at time frame t, to calculate entropy, the signal x(t) needs to be discretized over a period of 30, matching the fps assumption used in this study. This process results in $x_{i}$ where i=1,2,…,30. Each $x_{i}$ covers a specific range that x(t) can take. The entropy can be calculated as follows:(10)$$H\left(s\right)=-\sum_{i=1}^{30}p\left(x_{i}\right)\log_{2}p(x_{i})$$
$where\;p(x_{i})$ represents the probability that $x\left(t\right)$ falls within the range specified by $x_{i}$. Higher entropy indicates a more complex or less predictable signal, while lower entropy suggests a more regular, predictable pattern.Zero crossings: Zero crossings are a count of how many times the signal crosses the horizontal axis—that is, changes from positive to negative or vice versa. This feature also gives some insight into the frequency content of the signal. Given that x(t) represents the signal amplitude at time frame t, zero crossings can be formulated as follows:(11)$$\begin{array}{cc} Z & =\sum_{i=1}^{n-1}I\left(t,\;t+1\right)\;\;\\ I\left(t,\;t+1\right) & =\left\{\begin{array}{c} 1,\;\;if\;x\left(t)\cdot x(t+1\right)<0\\ 0,\;\;if\;x\left(t)\cdot x(t+1\right)>0 \end{array}\right. \end{array}$$Spectral centroid and spectral bandwidth: The spectral centroid is that point, considering the signal in the frequency domain, where the signal can be said to balance domain while spectral bandwidth is a measure of the spread or width of the frequency spectrum around the spectral centroid. They provide a measure of the amount the energy that is concentrated or dispersed across frequencies. Given that X(f) represents the frequency domain of the signal obtained by implementing x(t) into DFT in Formula (2), spectral centroid and spectral bandwidth can be formulated as follows:(12)$$\begin{array}{l} SC=\frac{\sum_{f}f\cdot \left|X\left(f\right)\right|}{\sum_{f}\left|X\left(f\right)\right|}\\ SB=\sqrt{\frac{\sum_{f}{\left(f-SC\right)}^{2}\cdot \left|f(X)\right|}{\sum_{f}\left|X\left(f\right)\right|}} \end{array}$$Dominant frequency and total power: Dominant frequency is the frequency at which, in the Fourier spectrum of the signal, its magnitude is the largest, whereas total power is a notion of the overall energy, computed by summing over squared amplitudes. Given that X(f) is the representation of time-domain signal x(t), dominant frequency and total power can be calculated as follows:(13)$$\begin{array}{cc} f_{d} & =\arg\underset{f}{\max}\left|X\left(f\right)\right|\;\\ P_{total} & =\sum_{t}{\left|x\left(t\right)\right|}^{2} \end{array}$$

## 3. Experiments

### 3.1. DeepFace for Gender and Age Prediction

In this study, it is necessary to utilize a sensor to automatically predict the physical properties. Therefore, DeepFace, a popular open-source framework for facial analysis, is utilized. DeepFace, which internally uses deep learning models pre-trained on very large datasets, offers a very high degree of accuracy for various facial attribute predictions, including age estimation and gender classification. Currently, the library supports several backend models, such as VGG-Face, Google FaceNet, and OpenFace. The VGG-Face backend provided by DeepFace was used in our implementation to analyze the first frame of every video regarding predicted age and gender due to its balance of speed and accuracy.

### 3.2. Experiment with Random Forest and Support Vector Regression for Multimodal Heart Rate Calculation

In this section, we outline the workflow utilized for our experiment, which encompasses a series of systematic steps designed to ensure a comprehensive evaluation of the machine learning models for heart rate estimation. The process begins with RoI extraction. Following this, feature engineering is employed to extract meaningful features from the signal data, including critical metrics such as the maximum, minimum, and peak-to-peak values; skewness; and kurtosis, which are vital for accurate model performance. All the features collected from facial videos will be concatenated with physical features such as age, gender, weight, height, and BMI.

Next, we conduct model selection, opting for support vector regression (SVR) and random forest regression by tuning the hyperparameter. This tuning phase is executed using a grid search method, where various hyperparameters are evaluated to find the optimal configurations for each model. Hyperparameters for each model are given as follows:
Hyperparameters for SVR: Kernel ∈ {Linear, RBF}, $C\in \left\{0.01,\;0.1,\;1,\;10,\;100\right\},\;\gamma \in \left\{0.01,\;0.1,\;1,\;10,\;100\right\}$ and $\epsilon \in \left\{0.001,\;0.01,\;0.1,\;1,\;10\right\}$;Hyperparameters for Random Forest Regression: number of trees n∈{50, 100, 200}, max depth d∈{none, 5, 10, 20}, and min samples per split m∈{2, 5, 10, 20}.

All the processes are implemented using the scikit-learn library, which is a well-known and open-source tool for data analysis and machine learning implementation provided by Python.

Subsequently, we apply 5-fold cross-validation (CV), which is a statistical technique used to assess the performance of the tuned models on unseen data. This iterative process works by splitting the dataset into 5 folds, followed by training process using 4 folds as the training set and the remaining fold as the validation set. We repeat the training process until every fold becomes a training set. The result of the model can be obtained by averaging the results across all the folds [40]. Figure 6 below illustrates model evaluation using 5-fold CV.

Finally, we evaluate the models using performance metrics such as mean absolute error (MAE), root mean squared error (RMSE), and mean absolute percentage error (MAPE) to quantify their predictive accuracy. Some commonly used techniques for measuring model performance in this study are MAE, RMSE, and MAPE, which can be formulated as follows:(14)$$MAE=\sum_{i=1}^{n}\frac{\left|y_{i}-\hat{y_{i}}\right|}{n}$$
(15)$$RMSE=\sqrt{\sum_{i=1}^{n}\frac{{\left(y_{i}-\hat{y_{i}}\right)}^{2}}{n}}$$
(16)$$MAPE=\frac{100\%}{n}\sum_{i=1}^{n}\frac{\left|y_{i}-\hat{y_{i}}\right|}{y_{i}}$$
where $y_{i}$ is the actual value and $\hat{y_{i}}$ is the predicted value. For each metric, the lower the value, the better the performance of the model being tested. A detailed diagram illustrating this workflow is presented in Figure 7 to provide a visual representation of the steps taken throughout the experimental process.

### 3.3. Performance Results

This study explored machine learning integration and multimodal approach for heart rate measurement by combining two types of data: signal features obtained from the facial video as listed in Equation (8) until Equation (13) were given the name “Video” and physical attributes, such as age, gender, height, and weight, were given the name “Physical”. Two distinct machine learning models, namely support vector regression and random forest regression, are utilized. We systematically conducted ablation study by removing or altering specific video features and physical attributes to assess their impact on the MAE, RMSE, and MAPE. The focus was on understanding how each feature contributes to the model’s overall accuracy in predicting the target variable. Table 1 shows the performance of all the ablation experiments conducted in this study. The columns Video and Physical use checkmarks (✓) and crosses (✗) to indicate whether the corresponding input type is included in each method. A checkmark means the input is included, while a cross means it is not. Generally, the random forest models perform better than the SVR with the best-performing method obtained when using the multimodal approach with the lowest MAE, RMSE, and MAPE of 3.057, 10.532, and 4.2%, respectively. In contrast, the best SVR is obtained when using physical information only with the MAE, RMSE, and MAPE of 3.251, 12.797, and 4.9%, respectively.

To gain more valuable insights from the random forest model, we checked predictive performance and interpretability using feature importances as shown in Figure 8 Feature importance provides a quick sense of how each feature is useful for the decisions of the model and, therefore, helps identify the most influential variables driving the predictions. This visualization is useful for making informed decisions about which features to retain, prioritize, or analyze further. By focusing on the most influential features, we can not only improve model performance but also uncover actionable insights for the problem domain.

In addition, Figure 9 shows Shapley additive explanation (SHAP) values, which completely explain how each feature contributes toward the output of the model for specific predictions. The values to the right, or more positive SHAP values, increase the prediction, while the values to the left, or negative SHAP values, decrease the prediction. Features that are further away from zero have a greater influence on a particular prediction. The SHAP values tell not only the direction or magnitude of each feature’s effect but also consider the dependencies among the features, thus giving rich insight into the behavior of the model.

For deeper analysis, Figure 10 shows a partial dependence plot (PDP), which provides insights into the effect of various features on the predictions of the random forest model on the multimodal data. Each subplot shows the partial dependence of a single feature, revealing how changes in its values impact the model’s output. The y-axis, representing partial dependence, indicates the direction and strength of each feature’s influence on the prediction. Higher values suggest a positive effect, while lower values suggest a negative one. Meanwhile, the x-axis shows the range of feature values, which are often normalized to facilitate comparison across different features.

For certain features like zero crossings, peak to peak, and BMI, the PDP curves display notable variations across specific ranges, suggesting that these features significantly impact the model’s predictions in those areas. In contrast, features such as age, weight, height, skewness, and spectral bandwidth exhibit relatively flat lines, indicating that changes in these values have minimal effect on the prediction. The plot also shows the behavior of binary or categorical variables, like gender, where the flat line across 0 and 1 values suggests a limited impact on the outcome.

Features with sharp increases or decreases, like zero crossings, may indicate threshold effects, where a small change within a certain range can substantially affect predictions. This pattern suggests that the model may have learned non-linear relationships with these features, capturing complex interactions. Overall, this PDP grid highlights features such as zero crossings, peak to peak, and BMI which play a more prominent role in driving predictions, whereas others have less influence on the model’s decision-making process.

### 3.4. Comparison with State-of-the-Art Methods

To evaluate the performance of the proposed method, we compared the result of our work to four different state-of-the-art methods explained in the introduction. The comparison is carried out by applying the state-of-the-art methods to the dataset used in this study and evaluated using the same metrics. It aims to compare how the proposed methodology measures the heart rate much more accurately than the state-of-the-art. It is easy to see from Figure 11 that our work considerably outperforms the existing state-of-the-art methods of remote photoplethysmography in heart rate measurement, evident from the lowest MAE, RMSE, and MAPE of 3.06, 10.53, and 4.02%, respectively. The random forest with multimodal approach outperforms the random forest without multimodal by 30% in terms of error, indicating the multimodal approach improves the remote heart rate measurement. Furthermore, the integration of random forest model and multimodal approach outperforms the base POS with FFT by 72%, indicating that our approach is more reliable.

## 4. Discussion

In this study, we investigated the performance of machine learning integration and a multimodal approach for heart rate measurement by using the combination of the rPPG blood volume pulsatile signal extracted from the facial video and physical attributes such as age, gender, height, weight, and BMI. By combining these diverse data sources, we aim to improve heart rate estimation accuracy in non-contact scenarios, which is beneficial for telemedicine, fitness monitoring, and mass health screening.

This study explored machine learning integration and multimodal approach for heart rate measurement by combining two types of data: facial video as listed in Equation (8) until Equation (13) was given the name “Video” and physical attributes, such as age, gender, height, and weight, was given the name “Physical”. Two distinct machine learning models, namely support vector regression and random forest regression, were utilized. We systematically conducted an ablation study by removing or altering specific video features and physical attributes to assess their impact on the MAE, RMSE, and MAPE. The focus was on understanding how each feature contributes to the model’s overall accuracy in predicting the target variable. Figure 8 shows the performance of all the ablation experiments conducted in this study. Generally, the random forest models perform better than the SVR with the best-performing method obtained when using the multimodal approach with the lowest MAE, RMSE, and MAPE of 3.057, 10.532, and 4.2%, respectively. In contrast, the best SVR is obtained when using physical information only with the MAE, RMSE, and MAPE of 3.251, 12.797, and 4.9%, respectively.

Table 1 shows that the random forest regression with a multimodal approach is the best method which gives the lowest error with an MAE, RMSE, and MAPE of 3.057, 10.532, and 4.2%, respectively. This indicates the effectiveness of combining the two different data types in the prediction of heart rate. The models that did not include physical data or relied totally on video data had a higher error rate, underpinning the importance of physical attributes in enhancing the accuracy of the prediction.

In contrast, the SVR models perform comparably but are not better than the RF model that leverages both video and physical attributes. Interestingly, the SVR model with only physical data, Index 5, yielded a relatively low MAE of 3.251, while further addition of either video or physical features degraded its performance. This suggests that while the SVR is successful with physical data, it is sensitive to the integration of video features, possibly due to the feature complexity or noise contributed by the multiple data types, which is the main drawback of SVR as mentioned by Farahmand, Desa, and Nilashi [41].

From these findings, it becomes clear that a multimodal approach, especially one that incorporates video and physical information, could greatly affect the performance of the model in estimating heart rate. The combined-data RF model’s superior performance strengthens the promise of multimodal machine learning methods for improvement in the accuracy of remote heart rate measurement systems. This has led to the conclusion that multiple sources of data contribute to the overall better performance for applications, such as rPPG, when compared with single-modality methods.

These conclusions are further supported by the feature importance and SHAP value analyses of the best-performing RF scenario shown in Figure 9. First, it shows zero crossings as the most critical feature, with its importance score way higher than the other features, valued at 0.3979; hence, it contributed most to the model’s predictions. Physical features such as BMI and dominant frequency also contribute to key roles but at a lower degree compared to the others. The SHAP value plot in Figure 9 offers more granularity on this, pointing out that zero crossings tend to have a consistent, high positive impact on the model output, while the impact of features such as BMI and peak to peak varies depending on their value. The domination of zero crossings as the most important feature in this study is due to its ability to measure the frequency of estimation, which is correlated with the heart rate.

Coupled with the discussion of feature importance, this suggests that the better performance of the RF model is indeed derived from its ability to exploit both video-derived features such as zero crossings and physical attributes like BMI. Allowing it to capture the complex interaction between these multimodal features tends to yield a low error rate in heart rate prediction. The SHAP analysis points out how these key features weigh in the output from the model, further reinforcing the fact that the conclusion must be about the inclusion of both video and physical data for the best performance. These results align with our hypothesis that physical attributes such as BMI influence the heart rate prediction [19,20,21].

The comparison in Figure 11 shows that the combination of random forest regression, POS algorithm, and multimodal data, which includes both video and physical attributes, achieves the lowest error rates among all the methods evaluated, with an MAE of 3.057, RMSE of 10.532, and MAPE of 4.2%. These results significantly outperform the other state-of-the-art methods, such as Green + FFT, ICA + FFT, and CHROM + FFT, which show considerably higher error rates. For instance, the Green + FFT method exhibits an MAE of 15.217 and a MAPE of 18.5%, highlighting the vast improvement achieved by integrating multimodal data with the random forest model.

## 5. Limitations and Future Works

Despite the promising results, there are some limitations in this study. First, the dataset used in this work is relatively small because no open-source site provides facial videos along with physical attributes information. Due to limited access to a large multimodal dataset, there is a risk that the model may be overfitting to the dataset used in this study. Future work should prioritize the collection of a larger, more diverse dataset to improve the model’s generalizability and robustness across different populations.

Furthermore, an inaccurate physical attribute feed to the system such as the input of false weight or height may influence the result because measuring heart rate is based on false information. The limitation of this study is one that should be resolved in the future by collecting more multimodal datasets and aiming to develop a model that automatically predicts physical attributes to avoid manual inputting with the intent of avoiding feeding false information to the measurement system. Lastly, the rPPG signals extracted from the video are sensitive to lighting and environmental conditions, which could affect the accuracy of heart rate estimation in non-controlled settings. Variations in light sources, shadows, and even skin tone may introduce noise in the video data. Future work could involve improving robustness to such factors by implementing data normalization techniques or training the model with data captured under diverse lighting conditions.

## 6. Conclusions

In this study, we aim to make noninvasive heart rate measurements more robust and reliable. We propose a machine learning integration into multimodal data including facial video, age, gender, weight, height, and BMI. The plane orthogonal to skin algorithm is utilized to extract blood volume pulsatile signal from the RGB facial video. Region of interest for video recording covers facial areas without eyes, nose, and lips. Signal filtering is implemented to denoise the obtained signal followed by signal extraction. Several features are selected, such as the maximum, minimum, and peak-to-peak value; skewness; kurtosis; zero crossings; spectral centroid; spectral bandwidth; dominant frequency; total power; and entropy value of the signal. Two machine learning models, support vector regression (SVR) and random forest regression are compared for this task. The experimental results demonstrate that incorporating a multimodal approach enhances model performance, with the random forest regression model achieving the best result surpassing the state-of-the-art methods. Furthermore, our method improves the robustness of non-contact heart rate measurement.

## Figures and Tables

**Figure 1 sensors-24-07537-f001:**
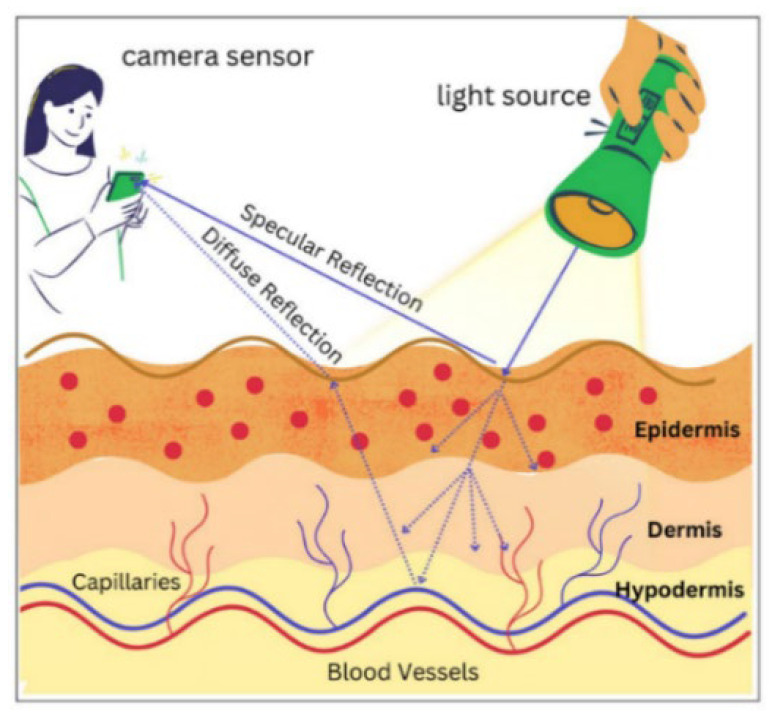
Illustration of remote photoplethysmography.

**Figure 2 sensors-24-07537-f002:**
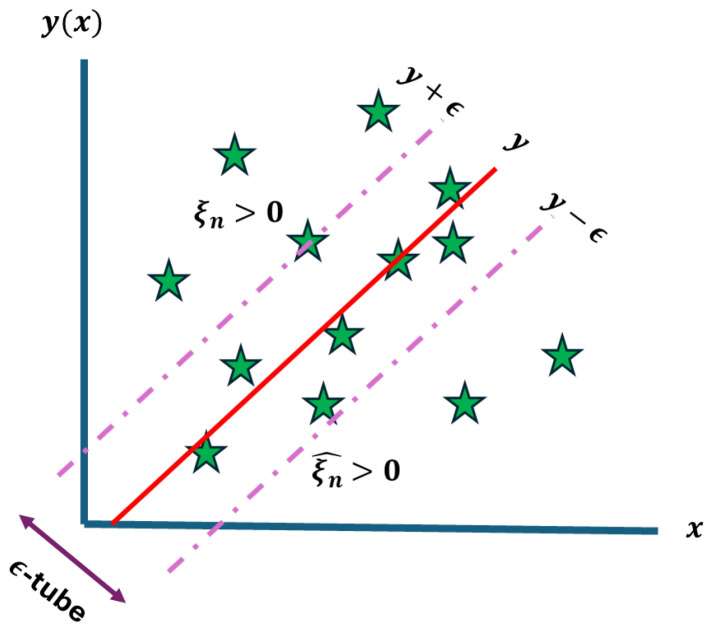
Hyperplane problem of the SVR.

**Figure 3 sensors-24-07537-f003:**
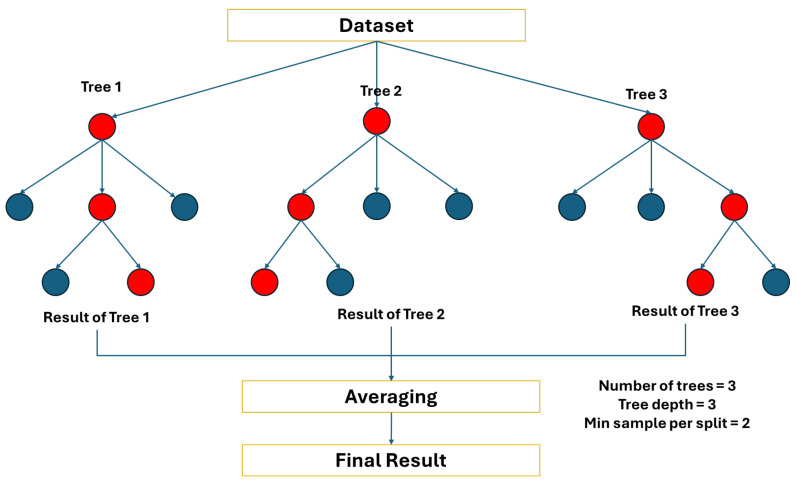
Random Forest Regression model.

**Figure 4 sensors-24-07537-f004:**
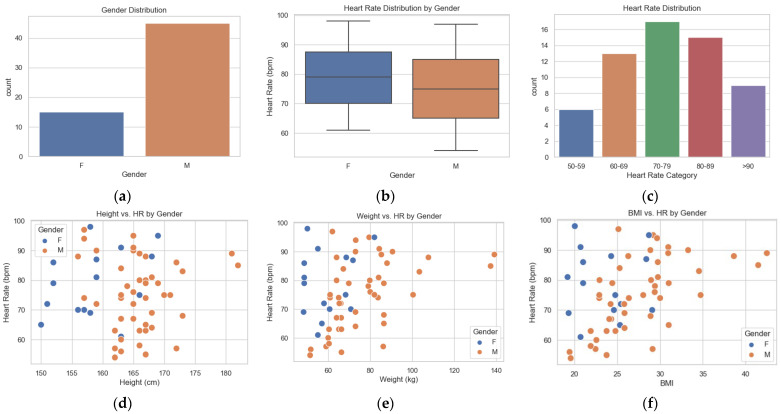
Dataset used in this study. (**a**) Gender distribution; (**b**) Heart rate distribution by gender; (**c**) Heart rate distribution by category; (**d**) Height vs. Heart Rate by Gender; (**e**) Weight vs. heart rate by gender; (**f**) BMI vs. heart rate by gender.

**Figure 5 sensors-24-07537-f005:**
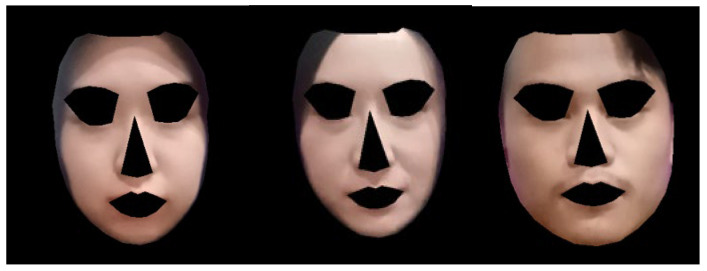
Region of interest used in this study.

**Figure 6 sensors-24-07537-f006:**
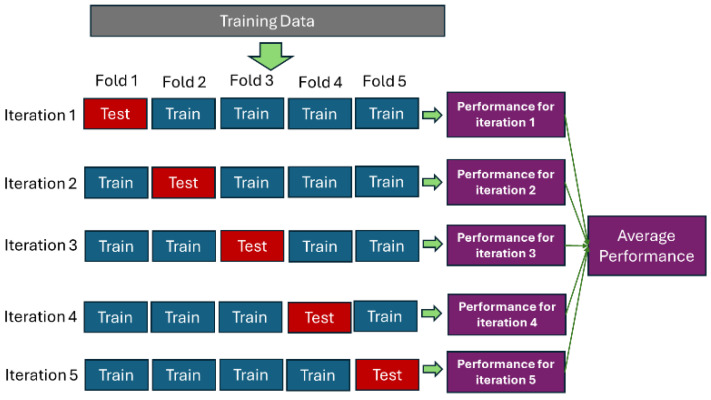
5-fold cross-validation.

**Figure 7 sensors-24-07537-f007:**
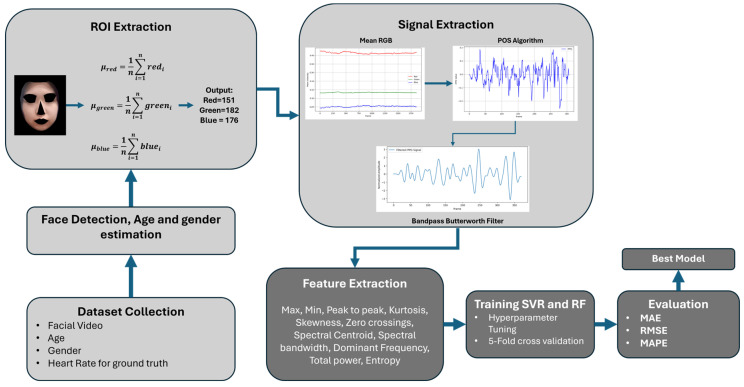
Workflow of the proposed method.

**Figure 8 sensors-24-07537-f008:**
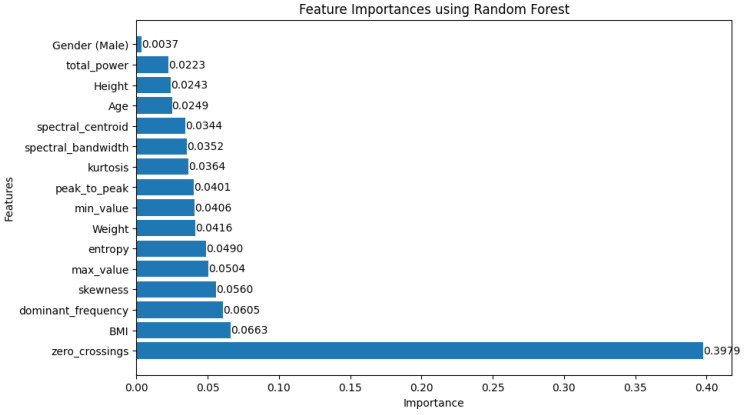
Feature importance.

**Figure 9 sensors-24-07537-f009:**
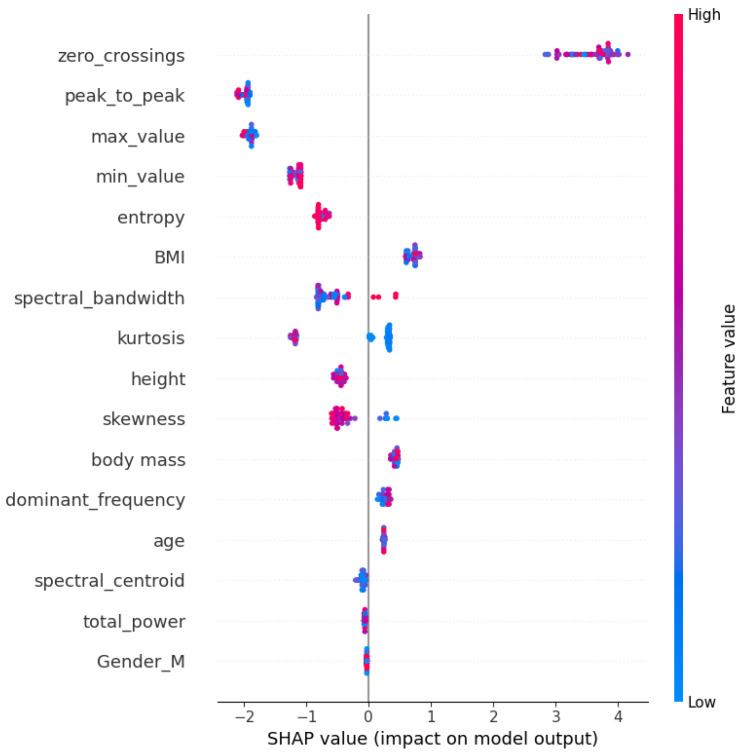
Shapley additive explanations.

**Figure 10 sensors-24-07537-f010:**
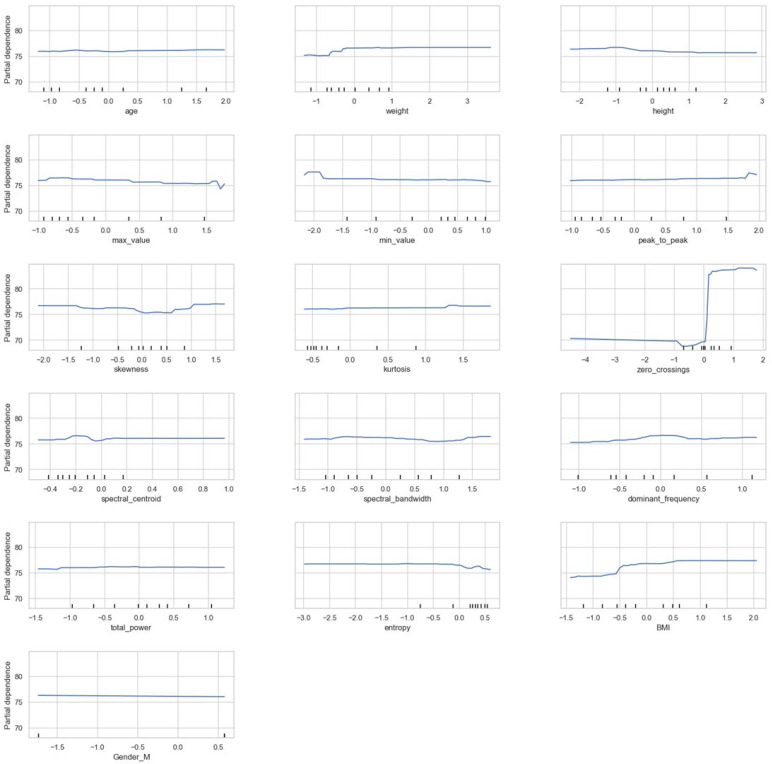
Partial dependence plot for random forest model.

**Figure 11 sensors-24-07537-f011:**
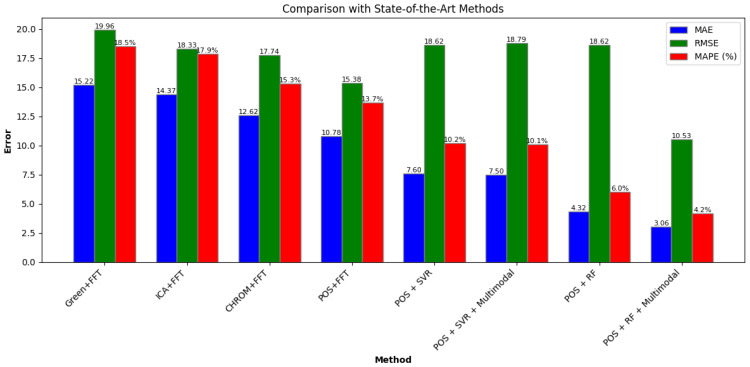
Comparison with state-of-the-art methods.

**Table 1 sensors-24-07537-t001:** Performance of proposed methods.

Index	Method	Video	Physical	MAE	RMSE	MAPE	Hyperparameter Pairs
1	RF	✓	✗	4.321	18.624	6%	n=70, d=7, m=None
2	RF	✗	✓	6.612	11.018	8.3%	n=120, d=5, m=2
3	RF	✓	✓	3.057	10.532	4.2%	n=80, d=7, m=2
4	SVM	✓	✗	7.601	18.624	10.2%	C=1000, ϵ=10, γ=0.01, kernel=linear
5	SVM	✗	✓	3.251	12.797	4.9%	C=1000, ϵ=0.001, γ=1,kernel=RBF
6	SVM	✓	✓	7.496	18.787	10.1%	C=1000, ϵ=10, γ=0.01, kernel=linear

## Data Availability

The data presented in this study are available upon request from the corresponding author due to privacy and ethical restrictions, as the data involve human subjects.

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
