# Peer review of "Integrating Remote Photoplethysmography and Machine Learning on Multimodal Dataset for Noninvasive Heart Rate Monitoring"

_sensors, 2024, doi:10.3390/s24237537_

Round 1

Reviewer 1 Report

Comments and Suggestions for Authors

The article clearly addresses a topic that is highly relevant to the general public, focusing on enhancing heart rate estimation accuracy in non-contact settings, which holds value for telemedicine, fitness tracking, and large-scale health screenings.

However, the authors need to more thoroughly address the study’s limitations. Currently, the paper lacks a dedicated section where the authors discuss the constraints and challenges of their research from their perspective.

Reviewer 2 Report

Comments and Suggestions for Authors

Dear Authors,

Thank you for the opportunity to review your manuscript titled "Integrating Remote Photoplethysmography and Machine Learning on Multimodal Dataset for Noninvasive Heart Rate Monitoring." I found your work on this important subject promising. However, I would like to offer the following suggestions to improve the clarity and overall quality of the manuscript:

Formula Presentation, clarify parameters and symbols consistently across the formulas for better readability and understanding.

Ensure proper formatting and spacing around operators to enhance legibility.

Include more intermediate steps for algorithms like SVR and Random Forest, which would aid readers in following your methodology.

Algorithm Explanation:

The steps of algorithms, such as the Plane-Orthogonal-to-Skin (POS), need clearer explanations, including assumptions, inputs, and outputs.

Strengthen the logical flow between equations and the surrounding text for greater coherence.

English and Grammar. There are several grammatical issues throughout the manuscript, and some sentences are overly long or awkwardly phrased. Refining these will greatly improve readability.

Simplify technical jargon and clarify terms, especially for concepts like "cross-validation" and "modalities" to ensure broader accessibility.

Comparison, the differences in performance between your proposed method and benchmark methods should be more explicitly highlighted.

I recommend adding more visual aids (e.g., graphs, tables) to compare the performance metrics more clearly.

I believe these changes will help improve the clarity, accuracy, and presentation of your work. I look forward to seeing how these revisions enhance your valuable research.

Best regards

Comments on the Quality of English Language

Fair

Reviewer 3 Report

Comments and Suggestions for Authors

Major issues:

  1. Making the research more reliable and applicable, I would suggest to collect large datasets based on different age, gender, and skin color. Also, the algorithm should be improved to reduce the effects of the light with advanced filtering or to adjust for lighting variations.
  2. I would suggest using models or sensors to automatically estimate physical properties to keep data more accurate, thereby reducing errors avoiding manual input.
  3. Privacy and ethics are crucial when it comes to non-contact heart rate monitors. Therefore, the user's privacy and data protection must be a major concern in the case of video promotion. I suggest clarifying this issue.
  4. The resolution used here is 30 frames per second and 480 × 640 resolution, which may affect performance on different devices and scenarios. I would recommend to enhance the resolution.
  5. This study used only two modalities (facial video and physical features). Other important physical data, such as skin tone or different facial expressions, were not included, which could have added more variety and accuracy to the results. I would recommend to consider these issues.

Minor issues:

  1. More comparative studies are needed to be discussed in this field.
  2. No explanation is given about voice signal frequency analysis.
  3. Although SHAP value and feature importance were used, more in-depth analysis, such as partial dependence plots (PDP), would have yielded more nuanced information about the performance and impact of features.
  4. Model evolution and accuracy comparison of their related different model visualizations are needed to be enhanced.
  5. The limitations of this model and where it can be used in the future are not well-explained.
